# A scientometric analysis of fairness in health AI literature

Isabelle Rose I. Alberto[1], Nicole Rose I. Alberto[1], Yuksel Altinel[2], Sarah Blacker[3], William Warr Binotti[4], Leo Anthony Celi[5,6,7], Tiffany Chua[8], Amelia Fiske[9], Molly Griffin[6], Gulce Karaca[10], Nkiruka Mokolo[11], David Kojo N Naawu[11], Jonathan Patscheider[12]*, Anton Petushkov[13], Justin Michael Quion[14]*, Charles Senteio[15], Simon Taisbak[16], İsmail Tırnova[17], Harumi Tokashiki[18], Adrian Velasquez[18,19], Antonio Yaghy[20], Keagan Yap[21]

1 University of the Philippines College of Medicine, Manila, Philippines, 2 Bagcilar Research and Training Hospital, General Surgery Department, University of Health Sciences, Istanbul, Turkey, 3 Department of Social Science, York University, Toronto, Ontario, Canada, 4 New England Eye Center, Tufts Medical Center, Boston, Massachusetts, United States of America, 5 Institute for Medical Engineering and Science, Massachusetts Institute of Technology, Cambridge, Massachusetts, United States of America, 6 Department of Medicine, Beth Israel Deaconess Medical Center, Boston, Massachusetts, United States of America, 7 Department of Biostatistics, Harvard T.H. Chan School of Public Health, Boston, Massachusetts, United States of America, 8 University of San Francisco, San Francisco, California, United States of America, 9 Institute for History and Ethics in Medicine, School of Medicine, Technical University of Munich, Munich, Germany, 10 Department of Medicine, Massachusetts General Hospital, Boston, Massachusetts, United States of America, 11 Meharry Medical College School of Medicine, Nashville, Tennessee, United States of America, 12 Trust Stamp Denmark, Copenhagen, Denmark, 13 University of Michigan, Ann Arbor, Michigan, United States of America, 14 University of the East Ramon Magsaysay Memorial Medical Center Inc, Quezon City, Philippines, 15 Department of Library and Information Science, Rutgers University School of Communication and Information, New Brunswick, New Jersey, United States of America, 16 Inviso by Devoteam, Aarhus, Denmark, 17 Department of General Surgery, Baskent University School of Medicine, Istanbul, Turkey, 18 Department of Medicine, Carney Hospital, Boston, Massachusetts, United States of America, 19 Warren Alpert School of Medicine at Brown University, Providence, Rhode Island, United States of America, 20 New England Eye Center, Boston, Massachusetts, United States of America, 21 Harvard College, Cambridge, Massachusetts, United States of America

055 These authors contributed equally to this work.
* jpatscheider@truststamp.net (JP); justinmicq@gmail.com (JMQ)

**Data Availability Statement:** All relevant data are within the manuscript and its Supporting Information files. All data is third-party data and does not require any special privileges to access.

## Abstract

Artificial intelligence (AI) and machine learning are central components of today's medical environment. The fairness of AI, i.e. the ability of AI to be free from bias, has repeatedly come into question. This study investigates the diversity of members of academia whose scholarship poses questions about the fairness of AI. The articles that combine the topics of fairness, artificial intelligence, and medicine were selected from Pubmed, Google Scholar, and Embase using keywords. Eligibility and data extraction from the articles were done manually and cross-checked by another author for accuracy. Articles were selected for further analysis, cleaned, and organized in Microsoft Excel; spatial diagrams were generated using Public Tableau. Additional graphs were generated using Matplotlib and Seaborn. Linear and logistic regressions were conducted using Python to measure the relationship between funding status, number of citations, and the gender demographics of the authorship team. We identified 375 eligible publications, including research and review articles concerning AI and fairness in healthcare. Analysis of the bibliographic data revealed that there is an overrepresentation of authors that are white, male, and are from high-income

More data that supports the findings of this study are publicly available here: https://public.tableau.com/app/profile/jonathan6077/viz/IstheFairnessCommuntyFair/IstheFairnessComminutyFair?publish=yes&fbclid=IwAR0_l5_b-lWLr_baGhCdwBKivVjyzVJg7CHG971EOEMiea2MSp33NXExVM The dataset can also be accessed at https://dataverse.harvard.edu/dataset.xhtml?persistentId=doi:10.7910/DVN/J1RTFG# Data and code repository can be found at https://github.com/anpetushkov/fairness-community.

**Funding:** The authors received no specific funding for this work.

**Competing interests:** The authors have declared that no competing interests exist.

countries, especially in the roles of first and last author. Additionally, analysis showed that papers whose authors are based in higher-income countries were more likely to be cited more often and published in higher impact journals. These findings highlight the lack of diversity among the authors in the AI fairness community whose work gains the largest readership, potentially compromising the very impartiality that the AI fairness community is working towards.

## Introduction

The fields of medicine and technology are undeniably intertwined; progress in one field often drives innovation in the other. It is no surprise that artificial intelligence (AI) is making headlines with its promise to inform or even automate clinical decision-making. However, the greatest impact this innovation has is not the language models trained on billions of parameters nor the generative models that create images from text prompts. Rather, complex bias exists within the data and it takes form in various ways, ranging from outcomes that are inconsistent across demographics, to subconsciously tainted tests and treatment decisions, and to influencing local clinical practice patterns in the form of institutional bias [1].

The concept of algorithm fairness has not only gained traction in the fields of artificial intelligence and software engineering, but also at global institutions such as the European Union. According to the European Commission, before people and societies develop, deploy, and use AI it first must be considered trustworthy. One of the critical elements of a trustworthy AI is fairness [2]. The European Commission defines it as having two dimensions: the "equal and just distribution of benefit and cost" to ensure freedom from unfair bias, discrimination, and stigmatization; and "the ability to contest and seek effective redress against decisions made by AI systems and by the humans operating them."

Over the last five years, unfairness in machine learning has gone from virtually unknown to often making headlines. Significant instances of undesirable bias induced in automated processes are frequently identified. One in particular occurred in 2016, following an article by ProPublica, an independent nonprofit news organization focusing on accountability, justice, and safety. It revealed that a software used by judicial courts across the United States was discriminating against Black and Hispanic prisoners during parole hearings [3]. In the same vein, machine learning models applied in healthcare are equally prone to similar issues. In 2019, it was revealed that millions of black patients had been misdiagnosed as a result of racial imbalances in a health algorithm used to triage patients. Consequently, the presence of biases and discrimination in machine learning models used in healthcare fosters the risk of misdiagnosis for certain demographic population groups, ultimately leading to loss of life [4].

Definitions and metrics of fairness in medical algorithms subsequently appeared in the medical literature. However, it should be noted that the meaning of fairness also extends to researchers involved in publishing datasets and studying biases. It is prudent to consider the bias woven into the very fabric of the algorithm itself that reflects the human assumptions of those who created it. This is a problem of representation and exclusion, and of epistemic narrowing that can lead to the perpetuation of structural inequities. Through her concept of "strong objectivity," the Science and Technology Studies scholar Sandra Harding has shown that the exclusion of marginalized authors, including People of Color, scholars based in low-income countries, and women, among others, from research and publishing is not only unjust, but also diminishes the scientific knowledge produced. In order to attain a stronger version of

scientific objectivity, and to create a science that can work towards equity and justice, Harding argues that we need to fortify that science by increasing the diversity of academic authorship as much as possible [5]. Which leads to the crux of the matter: how diverse is the fairness of the AI community proposing these definitions and metrics of fairness? It is important to bring attention to the diversity, or lack thereof, within this community to help prevent future propagation of bias and promote equity in health care.

## Methods

### Search strategy

In order to analyze the AI fairness community in healthcare, an in-depth descriptive study was done measuring aspects related to the publications in the field with a bibliometric review. The AI field was defined with terms that included machine learning, deep learning, convolution neural network, and natural language processing. Fairness overlapped in terms of health equity and health disparities.

Literature search was conducted in December 2022 using PubMed, Google Scholar, and Embase. These databases were chosen for their popularity, author familiarity, and ease of use with Python. The collected data were manually curated to secure the field of interest.

The search was conducted with the help of librarian Paul Bain, Ph.D., MLIS, from Harvard Medical School's Countway Library [S1 Appendix].

### Eligibility of articles

Studies were considered eligible for inclusion if they met the following criteria: (i) Does the paper discuss machine learning fairness? (ii) Is the paper related to healthcare? and (iii) does it discuss clinical applications?

If the three questions were affirmative, then the article was eligible. If a paper's eligibility was still uncertain, it was cross-checked by another author.

As scope of the analysis was to map the gender and ethnic representation in the community of AI fairness within healthcare, scientific and non-scientific articles were screened for eligibility. The authors used manual vetting to narrow down the list of articles by reading the abstracts or full texts when the abstracts did not provide enough information. Out of the 1614 articles initially found through the search, 375 (23%) were determined to be eligible for further analysis, with a total number of authors of 1984.

### Data items

The bibliometric data obtained from Embase, PubMed, and Google Scholar provided the authors' first and last names, gender, race, article title, abstract, keywords, and URL. The articles were vetted manually and with Python package PyPaperBot 1.2.2. [6] to obtain enough bibliometric data and to ensure a thorough mapping and measurement of academic trends. For each eligible study, the following data were extracted: type of article–opinion or research, which country the paper originated from, the journal it was published in, publication year, number of times each article was cited, whether funding was provided, and the name of the funding organization if provided. Additionally, the originating countries were classified based on the World bank classification for income to the following: low-income ($<$1.0 USD per year), lower-middle-income (1.0–4.1 USD per year), upper-middle income (4.1–12.7 USD per year), and high income ($>$12.7 USD per year) (cf. GDP per capita (current US$) | Data (worldbank.org)) [7].

### Approach to identifying each author's nationality, race, and sex

To ensure consistency in our dataset and perform statistical analysis, we used pre-defined groups provided by search platforms to classify the gender and race/ethnicity of authors. Race and ethnicity was classified as White, Asian, Black, Hispanic, or 'none', by the search platforms, while Gender was recorded as male, female or none.

When collecting data on the author's gender, race, and ethnicity, the study relied on a variety of sources, including self-identification in terms of ethnicity and race, and the author's chosen pronouns. If this information was not available, information found on web pages and articles, and details related to the authors' affiliations or memberships in social or support groups, was used to determine gender, ethnicity and race. If this information was still unclear, the authors' gender, race, and ethnicity were determined based on photographs found on multiple websites including university websites, private web pages, YouTube, and social media platforms such as LinkedIn. To maintain the accuracy and validity of the data, the study cross-checked every author's gender, race, and ethnicity, against multiple sources of information, and in addition, each article and its inherent information were verified by another author of the paper to ensure the validity, consistency, and accuracy in the data collection process. When collecting data on the authors' countries of origin, the study went back as far as possible on the authors' past, reviewing information available on LinkedIn or faculty and research web pages. If the authors did not disclose their home country, the study considered the country of their furthest educational background. The country where authors are currently based was also investigated via the location of the author's affiliated institution listed in the article.

The approach used to identify race, ethnicity, and gender has its limitations. When analyzing the bibliometric data, the collected information on the author's race, ethnicity, and gender was found to be unreliable and inconsistent, which reflects the inherent complexity of the topic at hand. Not all of the websites used as sources in this study allowed for authors to state their own identity. As a result, not all information was equally accessible via web searches. Authors who identified as multiracial, nonbinary, or other situations where the data was unclear, were not included, as the pre-defined groups provided by search platforms did not account for this. Moreover, the social constructs that can vary significantly depending on their socio-political context, gender, ethnicity and race, cannot and should not be directly determined from a picture. This means that in some cases, the information found may not completely reflect the author's preferred identities, which highlights some of the methodological challenges and the complexities of this kind of intersectional demographic data gathering, and the difficulty of analyzing data regarding race, ethnicity, and gender on a large scale using quantitative methods.

Lastly, the income level of the author's country of origin (cf. GDP per capita (current US$) | data (worldbank.org)) [3] their affiliated institution, and whether it is a minority-serving institution (cf. MSI List 2021.pdf (rutgers.edu), and their highest academic degree obtained (MD, Ph.D., etc.) [8] were investigated. To confirm the statistical certainty of the paper, the accuracy of the author's review of race, ethnicity, gender, country of origin, income level, etc. for each article's datapoint was verified multiple times by other authors involved in the study. Emphasis is placed on the first and last author due to their significant roles during the research process.

Nonetheless, this approach was chosen, because such determinations are difficult to make without engaging with a more in-depth survey of all authors to accurately record their preferred race, ethnicity, and gender.

Research on diversity requires a high level of reflexivity, including reflecting on one's own positionality in relation to matters of fairness in research. As such, we would like to situate ourselves in relation to this scholarship. Among the authors on this paper, 12 identify as male and

10 identify as female and 0 identify as non-binary. In terms of ethnicity, 10 authors identify as White, 8 identify as Asian, 3, identify as Black and 1 identifies as Hispanic. The authors come from the following countries of origin: USA (9), Turkey (2), Philippines (2), Canada (2), Denmark (2), Germany (1), Brazil (1), Lebanon(1), Peru (1). The idea for this research was inspired by conversations we have had with others in the field on matters of race, gender, and representation within AI and academia, and our own experiences of relative privileges working within this system.

## Statistical analysis

Regression analyses were performed to evaluate multiple factors that influence the number of citations and the presence of funding during the study. Only 253 papers identified as research papers were included in this analysis as opinion pieces generally have little direct funding nor are widely cited beyond the community.

## Results

Bibliographic data was directly obtained from Embase, which yielded 242 articles. Data from the PubMed API yielded a total of 875 articles. Finally, another 497 articles from Google Scholar were found using the package PyPaperBot 1.2.2 [6] in Python 3.9.12. In total, 1614 articles potentially related to AI fairness were collected.

Data was cleaned in Microsoft Excel. Spatial diagrams were generated using Public Tableau. Base map layers obtained from OpenStreetMap and go-cart.io. Additional graphs were generated using Matplotlib 3.5.1 [9] and Seaborn 0.11.2 [10]. The linear and logistic regressions were analyzed using the Python package statsmodels 0.13.2, and the t-tests were analyzed using the Python package scipy 1.7.3.

## Distribution of each author's ethnicity and gender

The results showed that, overall, 794 (40.0%) of the authors were female, and 1190 (60.0%) were male (S1 Appendix). When looking specifically at the first and last authors, 155 (41.3%) of the first authors were female, and 110 (32.2%) of the last authors were female (Fig 1).

When the author's race distribution was analyzed, the study categorized the race of 1966 authors out of the total of 1984 authors in our curated database. The study found that the majority of the authors were White (1270; 64.0%), followed by Asian (533; 26.9%), Black (89; 4.5%), and Hispanic (74; 3.7%) (Fig 2).

When dividing the authors into two groups, whites and non-whites, the study found that among the first authors, 234 (62.4%) were white and 141 (37.6%) were non-white. Among the last authors, 251 (66.9%) were white, and 124 (33.1%) were non-white (S1 Appendix).

## Distribution of each nationality

When looking at the country of origin, it was clear that most articles were from the USA. The total author nationality distribution showed that 986 (49.7%) were from the USA, 142 (7.2%) were from Canada, 117 (5.9%) were from the UK, 115 (5.8%) were from China, and 83 (4.2%) were from India (Fig 3). From income levels, 1631 authors (82.2%) were from high-income countries, 209 (10.5%) were from upper-middle-income countries, 135 (6.8%) were from lower-middle-income countries, and 9 (0.5%) were from low-income countries.

When looking at the 375 first authors, 175 (46.7%) were from the USA, 27 (7.2%) were from Canada, 22 (5.9%) were from the UK, and 21 (5.60%) were from China (Fig 4A). Among the last authors, 179 (50.7%) were from the USA, 29 (8.2%) were from Canada, 20 (5.7%) were

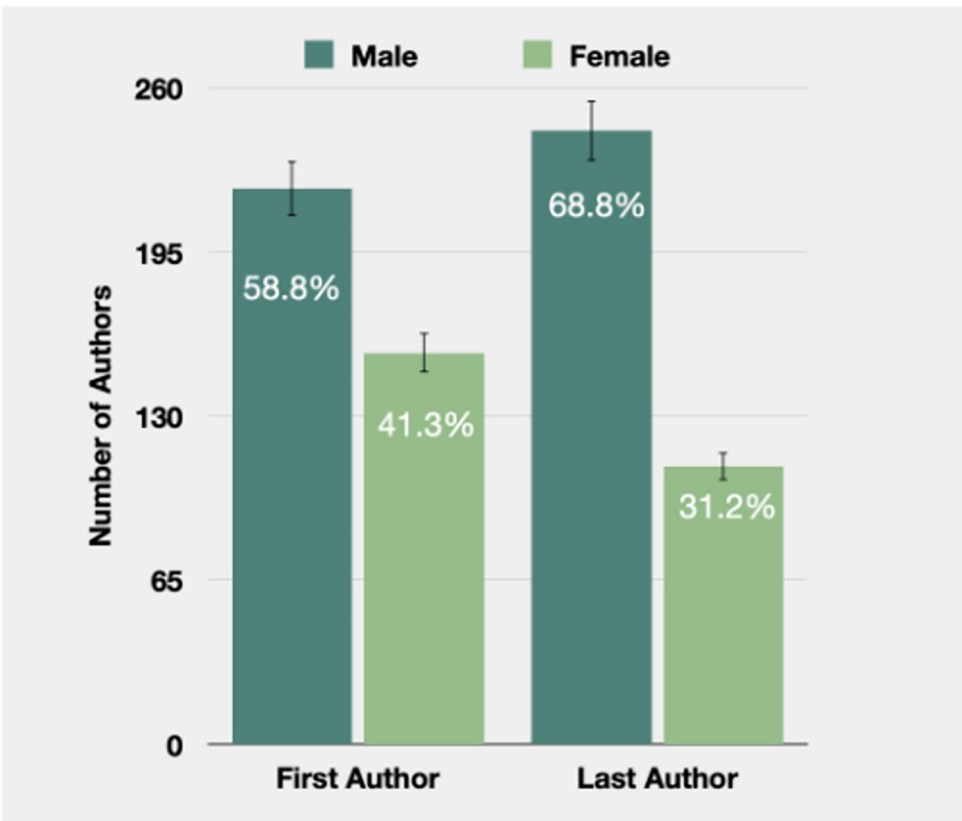

**Fig 1. Distribution of first and last author sex.**

from the UK, and 14 (4.0%) were from China (Fig 4A). For the first authors, 318 (84.8%) were from high-income countries, 32 (8.5%) were from upper-middle-income countries, 24 (6.4%) were from lower-middle-income countries, and 1 (0.3%) were from low-income countries.

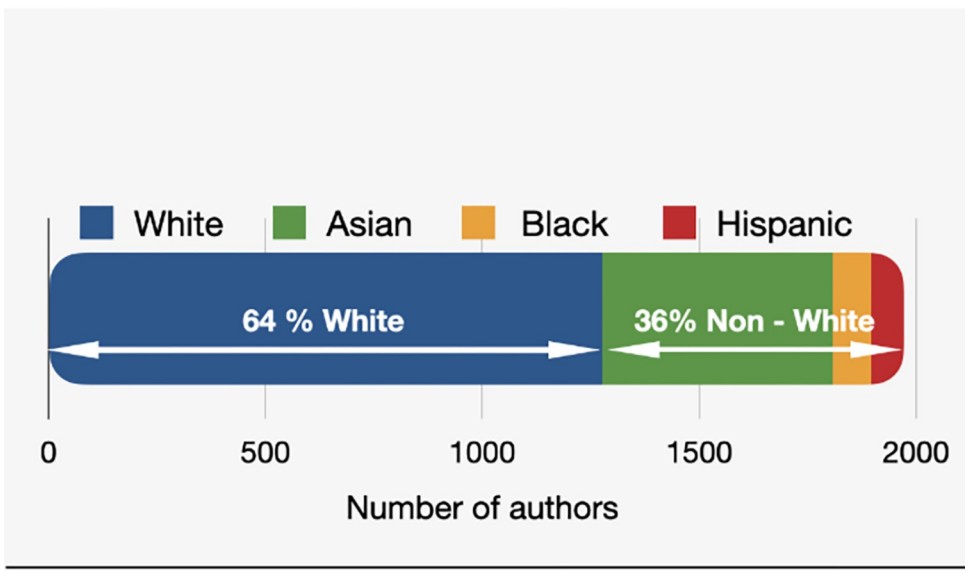

**Fig 2. Distribution of first, last, and all authors' race.**

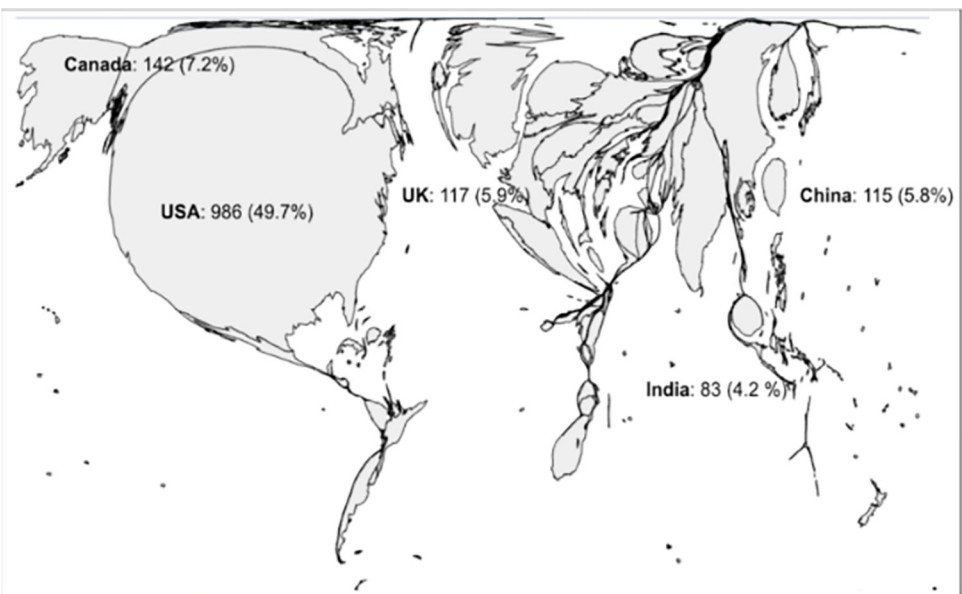

**Fig 3. Distribution of each nationality.** Base map layer data found at https://go-cart.io/cartogram, courtesy of go-cart. io, and is available under Creative Commons License CC-BY, found here:https://go-cart.io/faq.

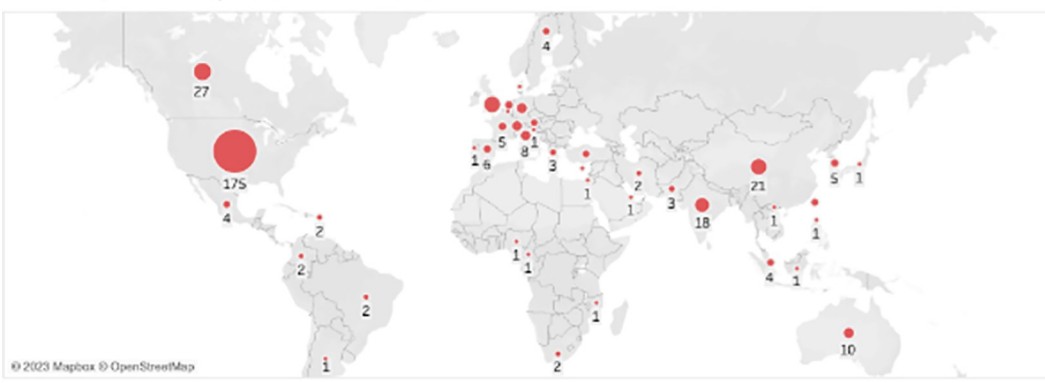

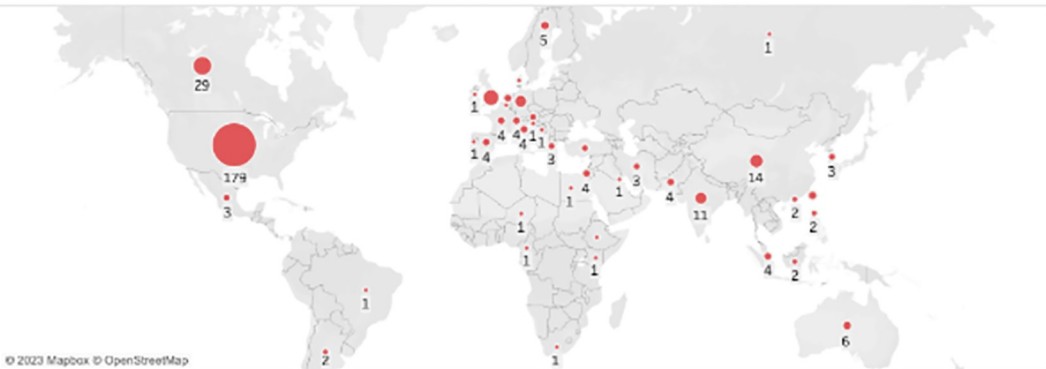

**Fig 4. Global dispersion of first and last author countries.** Base map found at https://www.openstreetmap.org/#map=2/43. 8/3.2 and data from OpenStreetMap and OpenStreetMap Foundation. Contains information from OpenStreetMap and OpenStreetMap Foundation, which is made available under the Open Database License, found here: https://www. openstreetmap.org/copyright.

For the last authors, 312 (88.4%) were from high-income countries, 27 (6.8%) were from upper-middle-income countries, 15 (4.2%) were from lower-middle-income countries, and 2 (0.6%) were from low-income countries (Fig 4B).

Further analysis was done to investigate where first and last authors are currently based. This was determined by the location of the institution listed in the article. Of the listed first authors, 347 of the 357 (92.5%) are currently based in institutions located in high-income countries. Of these, 220 (59%) are based in the US, 28 (7%) based in Canada, 24 (6%) based in the UK, 11 (3%) based in Australia, and 10 (3%) are based in Germany. There is only 1 first author (0.3%) based in a low-income country, Mozambique, and 10 (2.7%) based in lower-middle-income countries. This trend continues for the last authors. 332 (94%) of last authors are based in high-income countries at similar proportions with regards to country: 214 (61%) based in the USA, 31 (9%) based in Canada, 20 (6%) based in the UK, 11 (3%) based in Australia, and 8 (2%) based in Germany. Of the remaining last authors, 14 (4%) are based in upper-middle-income countries, 7 (2%) are based in lower-middle-income countries, and 0 are based in low-income countries.

## Citations and funding

By investigating citations across gender and ethnicity, it was observed that there was an over-representation of white-male authors. The data indicates, as illustrated in Fig 5, that papers with more male and white authors tended to receive more citations than those with fewer male and white authors.

On further investigation, it became evident that on average for both first and last authors, non-white and female authors receive fewer citations than white male authors, which is illustrated for last authors in Fig 6. From here, the data revealed that articles with male last authors accounted for a substantial 76.4% of all citations, with white male last authors alone responsible for 58.3% of the total citations for all articles (Fig 6).

The findings from Figs 5 and 6, prompts the argument that male authorship, particularly that of white males, may be associated with higher-impact articles published in high-impact journals.

The analyses in the study also suggest that higher-income countries may have a higher likelihood of being funded and producing higher-impact articles in terms of citations (Fig 7). This

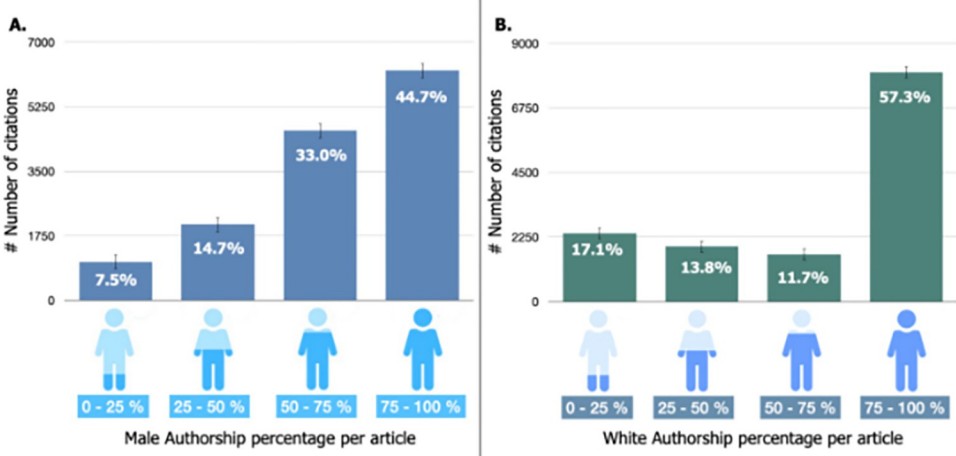

**Fig 5.** Distribution of Male Authorship (A) and White Authorship (B) per article with number of citations and percentage.

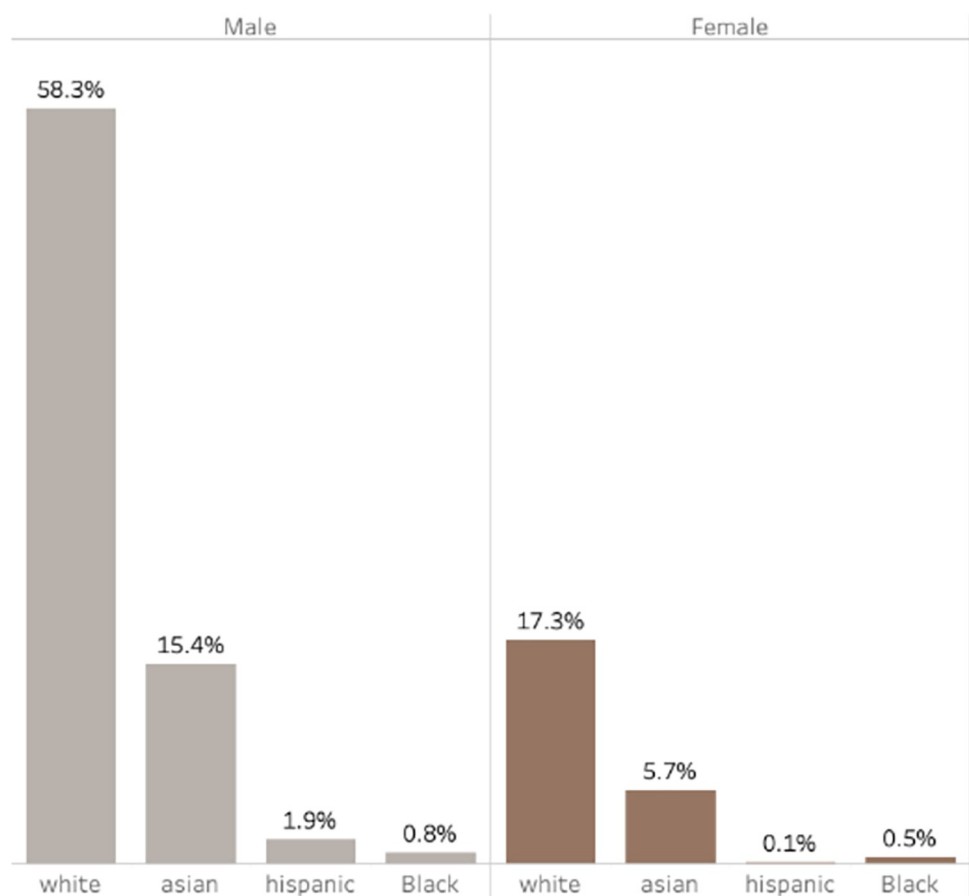

**Fig 6. Distribution of citations among last authors based on gender and ethnicity.**

could be due to higher-income countries having greater access to resources and funding, which could contribute to the production of higher-impact articles. Additionally, the research culture, infrastructure, and collaboration networks in higher-income countries may also play a role in producing impactful research. It is also important to consider the potential biases that may be present, such as language bias, publication bias, citation practices and funding. Recent scholarship on citational practices and politics draws attention to the ways that structural inequities among authors are reflected in citation practices, noting that scholars can take an active role in upending these hierarchies through an intentional transformation in their own citational practices [11, 12]. These biases could impact the analysis and interpretation of the results, and their access to funding, leading them to make less impactful articles.

Articles were also evaluated according to how often they get cited, in which year of publication was another variable in classification, which showed that more recent (closer to 2022) articles were most likely to be cited.

**OLS regression results (Citations).**   Regression analyses revealed that the percentage of female authors, percentage of white authors, race of authors, and the gender of authors are not significantly correlated with the likelihood of being cited. (Fig A, B in S1 Appendix) Publication year and the number of authors are the only factors affecting the number of citations.

**OLS regression results (Funding).**   Regression analyses revealed that the following parameters: percentage of female authors, percentage of white authors, gender of first and last

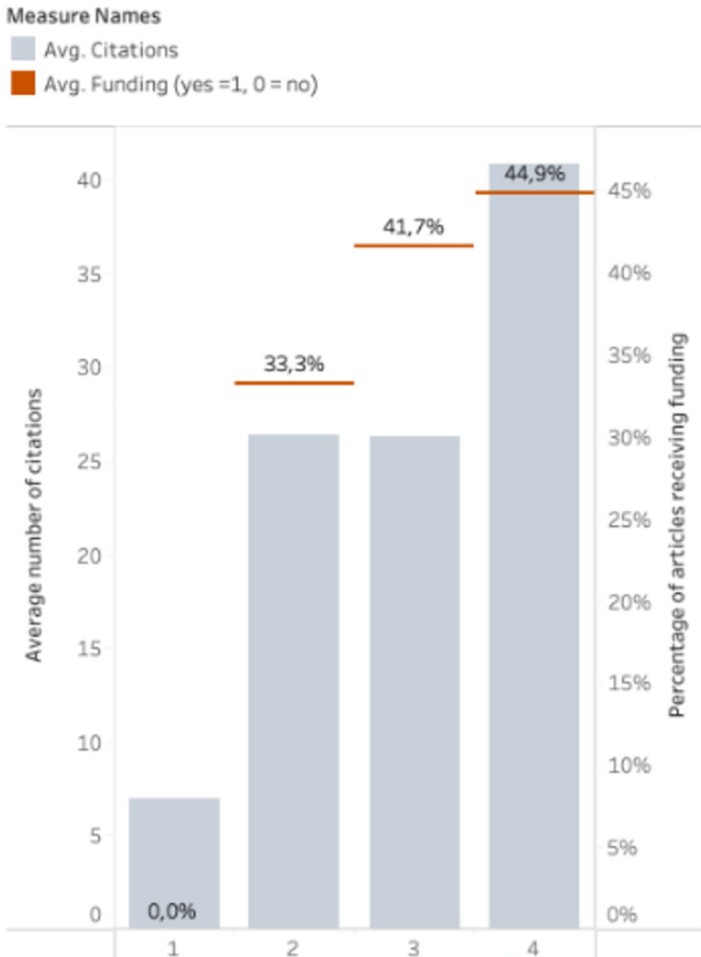

**Fig 7. Distribution of citations among last authors income level.**

authors, and first and last authors, whether white or non-white, do not affect funding. Additionally, the number of authors and years of publications does not affect being funded.

**Predictor factors of citation and funding.** Regression analyses revealed similar trends for citation and funding. The percentage of female authors, the percentage of white authors, and the gender and race of the first and last authors did not have a statistically significant effect on whether a paper was cited or whether it was funded. Instead, publication year was the only factor with a statistically significant effect on the number of citations a paper received.

## Discussion

The performed bibliometric study highlights the relative homogeneity of the authors of the AI fairness community, most notably seen in the distribution of gender, ethnicity, and countries of authorship. Male authorship, particularly that of white males, may be associated with higher-impact articles published in high-impact journals.

The gender and racial disparities among authors in academic publications are evident, with a notable overrepresentation of white male authors, especially in the positions of first and last authors. Among all authors, 60% were male, and this proportion increased to 69% when considering the distribution of first authors and last authors, as illustrated in Fig 1.

Furthermore, there is a clear relationship between the gender and ethnicity of authors and the impact of their publications, as demonstrated in Figs 5 and 6. Publications with a higher proportion of male authors, particularly those who are white males, tend to have a greater number of citations, indicating a higher impact. Notably, articles with male authors in the last author position accounted for a substantial 76.4% of all citations. White male authors, in particular, were responsible for 58.3% of the total citations across all articles. This is important as the role of the last author typically goes to the lead principal investigator of the team who supervises the project.

The increasing numbers of health disparities in underrepresented ethnic groups, and the underrepresentation of minority ethnic groups and women in academia has been well documented. Multiple studies and reports have addressed this trend and this study shows that it persists within the AI fairness community as well, where there is a significant disparity in gender and ethnicity in critical care medicine [13–20]. Women are underrepresented in leadership roles, which can limit their opportunities for publication and recognition [13–15], and are less likely to be first or last authors and less likely to be cited compared to male authors [14–16]. Certain racial/ethnic groups are represented only minimally, and Non-White groups–namely, Latinos, African-Americans, and American Indians–are underrepresented in health professions that require an undergraduate or graduate degree [17–20]. As the US demographic landscape undergoes rapid transformations and the recognition of growing health disparities within historically marginalized ethnic communities intensifies, the future of healthcare in the country will increasingly rely on a diverse healthcare workforce, as this can help improve cultural competence among health care professionals thus reducing health disparities [19, 20].

These studies align with the findings of this papers' analysis, as similar trends were observed. However, from the data, there was a slight indication that papers with female last authors were more likely to get a funding source, compared to the male counterpart, counterbalancing the notion of male predominance [19]. This finding highlights the positive impact that could be achieved, when equality measures are implemented by regulatory institutions in AI research within healthcare.

The analysis also shows that higher-income countries are more likely to produce higher-impact articles, most likely a reflection of the amount of funding received as demonstrated by Fig 7. Massuda et al show that underfunding a survey can lead to a significant reduction in quality, perpetuating the status quo [21]. This difference in funding perpetuates the current power dynamic where countries from underfunded institutions in low- and middle-income countries are less likely to produce high-quality research that is widely cited and well-regarded. Promoting research from underrepresented groups and communities is essential to promoting fairness and equity in research.

Authors are also more likely to be based in wealthier countries. This is understandable as skilled researchers from lower-income countries often relocate to higher-income countries for a variety of reasons, whether it is to conduct research or because they are offered scholarships. However, considering that 82% of authors are already from high-income countries, compared to the 0.5% of authors from low-income countries, this further underlines the lack of representation. The lack of last authors based in low-income countries is a glaring sign of the research being performed, as they play a large role in shaping the question and design.

## Possible actions

As progress is made in both AI and healthcare, equity and inclusivity must be prioritized as it can lead to more innovative and impactful research, and a science that works for all [22, 23].

Possible actions to ensure proper representation include supporting research capacity development in lower-income and lower-middle-income countries and promoting research

conducted by researchers of underrepresented gender identities and ethnic minorities. In a narrative review conducted by Bowsher and colleagues, several critical factors were identified to enhance research capacity in Low- and Middle-Income Countries (LMICs). These factors encompass the simultaneous addressing of individual, organizational, and institutional levels, ensuring sustainable funding and resource allocation, promoting capable and shared leadership within equitable partnerships, facilitating mentorship programs, establishing professional networks, and establishing links between research outcomes and policy/practice implementation. The process of strengthening research capacity necessitates focused investment, the implementation of mentorship initiatives, the cultivation of robust collaborations, and the implementation of effective monitoring and evaluation mechanisms. The success of capacity-building endeavors relies on long-term strategic planning and collaboration involving a diverse array of stakeholders, including researchers, program implementers, and policymakers across all levels [24].

By supporting this development, the global research landscape becomes more inclusive. This in turn helps to advance and strengthen medical knowledge and promote social justice within the scientific community. In addition, promoting collaboration and cooperation between researchers from diverse backgrounds and locations can also lead to more innovative and impactful research [22, 23].

Another way to promote diversity and inclusion in research is to establish guidelines for diversifying the composition of authors based on their ethnicity and sex. Providing formal training on equity issues and the importance of diversity in the research process can help educate researchers and promote greater awareness and understanding of these issues. This can be incorporated into the syllabi of academic institutions to ensure that the next generation of researchers is equipped with the knowledge and skills necessary to promote diversity and inclusion in their work. Diversity should be highlighted in published work and working groups. Disclosing authors' nationalities, races, ethnicities, and sexes can promote diversity and inclusivity. Inclusivity also begins at the door. Institutions should develop initiatives that can help to attract more diverse scholars through transforming institutional cultures and priorities, as well as recruitment, hiring, and promotion policies.

In addition to promoting diversity in the composition of working groups and authorship of published work, it is also important to consider diversity in the content of the work, for example, including diverse perspectives and experiences in the research or addressing issues that affect diverse communities. AlShebli et al. found that ethnic diversity had the strongest correlation with scientific impact [22]. Recruiters should always strive to encourage and promote ethnic diversity, be it by recruiting candidates who complement the ethnic composition of existing members, or by recruiting candidates with proven track records in collaborating with people of diverse ethnic backgrounds.

Researchers should seek to understand their own group composition and how it should coincide with the communities which the research may impact. Representativeness and collaboration with communities can result in better science [24] and as such, greater understanding and awareness of these groups' challenges and issues can ultimately lead to more effective solutions. It is also worth noting that groups with higher cognitive diversity are often more effective at complex problem-solving and can help to reduce biased judgment in strategic decision-making [24, 25].

Journals, editors, reviewers, and grantors can mandate that the author teams disclose their goals for achieving such diversity. Doing so would promote transparency and accountability and encourage authors to prioritize diversity and inclusion in their research. The National Institutes of Health (NIH) actively promotes diversity within the scientific community by encouraging conference grant applicants to include plans to enhance diversity in the selection

of organizing committees, speakers, other invited participants and attendees [26]. These plans will be assessed during the scientific and technical merit review of the application and will be considered in the overall impact score. The underrepresented groups include individuals from nationally underrepresented racial and ethnic groups, individuals with disabilities, individuals from disadvantaged backgrounds, and women. Encouraging authors to highlight their efforts to promote diversity in their groups can raise awareness of the importance of diversity and inclusion in the scientific field and promoting diversity and inclusion in all aspects of research can ensure that the work is more representative and relevant to a broader range of people, ultimately leading to more equitable and effective outcomes.

Although several significant publications resulted from PubMed and Google Scholar searches, some were excluded. We used the third-party package PyPaperBot when selecting papers resulting from Google Scholar searchers which enabled us to extract 497 papers out of potentially hundreds of thousands. A large portion of the articles was removed from further analysis through manual vetting. However, third-party APIs are the only ways to parse through Google Scholar results. PyPaperBot was used, but a limitation of all the APIs seen is that they can only fetch the first 1,000 results, even if there are more. The extent of the literature is more vast than what we were able to extract, and it is crucial to scale up this analysis to capture more of the literature base in the future. It should also be noted that this field is relatively new and rapidly changing. As such, many regional databases may not have been available or accessible at the time of the literature search.

While manually vetting and extracting the authors' demographic information, information may differ from the authors' preferred identities, specifically for gender, race and countries of origin, as race, ethnicity, and gender are social constructs [27] that can vary significantly depending on their socio-political context, and it may not accurately reflect the author's personal identity. Our analysis may have mischaracterized this vital information if their identities were not clearly stated on the internet. The use of predetermined categories for race and ethnicity made it particularly difficult to capture authors who may identify as multi-racial, or as belonging to several of these categories. It is also important to recognize that some people may not have the freedom or opportunity to publicly express how they identify. Similarly, assessments of whether an individual identified as non-binary were particularly challenging if not explicitly stated, and as such were not included in this study.

## Conclusion

Moving forward, it will be important to develop better methodologies for representing a diverse range of possible identifications in order to better study questions of diversity, and a preferred methodology would involve interviewing each author in order to accurately record their nationality, self-identified race, and sex, as well as expanding the categories, however due to the scale of the study, it was not possible to obtain self-identified information in all cases. Systemic changes that allow for proper expression of identification are also necessary. Despite the presence of some inaccuracy within the data, as a necessity to perform statistical analysis, the overall trends revealed within this data are clear.

As progress is made in both AI and healthcare, equity and inclusivity must be prioritized as it can lead to more innovative and impactful research, and a science that works for all. Thus the composition of the AI fairness research community is of the utmost importance as whether AI will be a tool which only those who meet certain criteria can benefit from or a platform that serves all communities no matter their demographics, depends heavily on those who have a say in its design.

## Supporting information

**S1 Appendix. Additional information and data.**
(PDF)

## Author Contributions

**Conceptualization:** Isabelle Rose I. Alberto, Nicole Rose I. Alberto, Yuksel Altinel, Sarah Blacker, William Warr Binotti, Leo Anthony Celi, Tiffany Chua, Amelia Fiske, Molly Griffin, Gulce Karaca, Nkiruka Mokolo, David Kojo N Naawu, Jonathan Patscheider, Anton Petushkov, Justin Michael Quion, Charles Senteio, Simon Taisbak, İsmail Tırnova, Harumi Tokashiki, Adrian Velasquez, Antonio Yaghy, Keagan Yap.

**Data curation:** Isabelle Rose I. Alberto, Nicole Rose I. Alberto, Yuksel Altinel, Sarah Blacker, William Warr Binotti, Leo Anthony Celi, Tiffany Chua, Amelia Fiske, Molly Griffin, Gulce Karaca, Nkiruka Mokolo, David Kojo N Naawu, Jonathan Patscheider, Anton Petushkov, Justin Michael Quion, Charles Senteio, Simon Taisbak, İsmail Tırnova, Harumi Tokashiki, Adrian Velasquez, Antonio Yaghy, Keagan Yap.

**Formal analysis:** Isabelle Rose I. Alberto, Nicole Rose I. Alberto, Yuksel Altinel, Sarah Blacker, Leo Anthony Celi, Tiffany Chua, Amelia Fiske, Molly Griffin, Gulce Karaca, Nkiruka Mokolo, David Kojo N Naawu, Jonathan Patscheider, Anton Petushkov, Justin Michael Quion, Charles Senteio, Simon Taisbak, İsmail Tırnova, Harumi Tokashiki, Adrian Velasquez, Antonio Yaghy, Keagan Yap.

**Writing – original draft:** Isabelle Rose I. Alberto, Nicole Rose I. Alberto, Yuksel Altinel, Sarah Blacker, William Warr Binotti, Leo Anthony Celi, Tiffany Chua, Amelia Fiske, Molly Griffin, Gulce Karaca, Nkiruka Mokolo, David Kojo N Naawu, Jonathan Patscheider, Anton Petushkov, Justin Michael Quion, Charles Senteio, Simon Taisbak, İsmail Tırnova, Harumi Tokashiki, Adrian Velasquez, Antonio Yaghy, Keagan Yap.

**Writing – review & editing:** Isabelle Rose I. Alberto, Nicole Rose I. Alberto, Yuksel Altinel, Sarah Blacker, William Warr Binotti, Leo Anthony Celi, Tiffany Chua, Amelia Fiske, Molly Griffin, Gulce Karaca, Nkiruka Mokolo, David Kojo N Naawu, Jonathan Patscheider, Anton Petushkov, Justin Michael Quion, Charles Senteio, Simon Taisbak, İsmail Tırnova, Harumi Tokashiki, Adrian Velasquez, Antonio Yaghy, Keagan Yap.

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
