## [Decision Letter · Decision Letter 0]

13 Sep 2023

PGPH-D-23-00817

A scientometric analysis of fairness in health AI literature

Dear Dr. Quion,

Thank you for submitting your manuscript to PLOS Global Public Health. After careful consideration, we feel that it has merit but does not fully meet PLOS Global Public Health’s publication criteria as it currently stands. Therefore, we invite you to submit a revised version of the manuscript that addresses the points raised during the review process.

We look forward to receiving your revised manuscript.

Kind regards,

Zahra Zeinali, MD MPH DrGH (c)

Academic Editor

Journal Requirements:

2. Please provide separate figure files in .tif or .eps format only and remove any figures embedded in your manuscript file. Please also ensure all files are under our size limit of 10MB.

3. We have noticed that you have a list of Supporting Information legends in your manuscript. However, there are no corresponding files uploaded to the submission. Please upload them as separate files with the item type 'Supporting Information'. 

4. Some material included in your submission may be copyrighted. According to PLOS’s copyright policy, authors who use figures or other material (e.g., graphics, clipart, maps) from another author or copyright holder must demonstrate or obtain permission to publish this material under the Creative Commons Attribution 4.0 International (CC BY 4.0) License used by PLOS journals. Please closely review the details of PLOS’s copyright requirements here: PLOS Licenses and Copyright. If you need to request permissions from a copyright holder, you may use PLOS's Copyright Content Permission form.

Potential Copyright Issues:

Figs 1-3: please (a) provide a direct link to the base layer of the map (i.e., the country or region border shape) and ensure this is also included in the figure legend; and (b) provide a link to the terms of use / license information for the base layer image or shapefile. We cannot publish proprietary or copyrighted maps (e.g. Google Maps, Mapquest) and the terms of use for your map base layer must be compatible with our CC-BY 4.0 license. 

"

Additional Editor Comments (if provided):

Reviewers' comments:

Reviewer's Responses to Questions

**Comments to the Author**

1. Does this manuscript meet PLOS Global Public Health’s publication criteria? Is the manuscript technically sound, and do the data support the conclusions? The manuscript must describe methodologically and ethically rigorous research with conclusions that are appropriately drawn based on the data presented.

Reviewer #1: Yes

Reviewer #2: Yes

2. Has the statistical analysis been performed appropriately and rigorously?

Reviewer #1: Yes

Reviewer #2: Yes

3. Have the authors made all data underlying the findings in their manuscript fully available (please refer to the Data Availability Statement at the start of the manuscript PDF file)?

Reviewer #1: Yes

Reviewer #2: Yes

4. Is the manuscript presented in an intelligible fashion and written in standard English?

Reviewer #1: Yes

Reviewer #2: Yes

5. Review Comments to the Author

Reviewer #1: This manuscript provides an in-depth bibliometric analysis of scholarly articles concerning the topic of artificial intelligence (AI) and medicine to further investigate the fairness of AI in literature. The authors reviewed 375 publications and reviewed corresponding multiple factors (including demographic data, funding, nationality, etc.) for each author.

With the rapid growth of AI in various fields, this article has a timely and relevant nature to it. I believe this article has the potential to contribute to the field and to PLOS Global Public Health with edits. The strength of this review lies largely in the analysis of these articles that yield significant results.

Attached please find additional comments and feedback, divided by section, which we believe can help strengthen your manuscript. We also advise that this manuscript be re-checked and read for grammatical errors.

Reviewer #2: This is an important area that fits in global Health issue. The authors used appropriate methodological approach to answer the review questions. Appropriate statistical tools and analysis were performed. However, it will be good if the authors consider the following points to revise and improve the manuscript.

ABSTRACT

1. Included papers. 375, were repeated in the methods and results. I suggest this is deleted from the methods in the abstract.

2. Authors should be consistent with the use of decimal place, I suggest 1 decimal place is used as it is largely used in the manuscript.

3. "The linear and logistic regressions were analyzed using Python"........ what was this analysis for?

4. The sentence before the last paragraph in their conclusion seem more like a finding than a conclusion. Authors can consider and revise.

5. "Most authors were from US, Canada and United Kingdom"....Can authors quantify what is referred to as "Most".

INTRODUCTION

6. Line 51 -54 may need revision. Long sentence and not clear.

7. It will of interest if authors expand on the fairness of AI in the introduction

METHODOLOGY

8. When was the literature search conducted? (period of search?)

9. How was papers excluded apart from the inclusion criteria, At what levels of were the papers screened?

10. How was the number of times a paper was cited identified? Authors should kindly elaborate

11. line 140.."authors who identified...." This aspect is not properly placed. I suggest it comes just after the inclusion criteria.

12. I suggest authors elaborate on the anaylsis procedure, as it stands, it does not reflect the content of the work

13. How many research papers were included in the regression analyses?

14. Line 190-196 ("As scope of the analysis.....") seem wrongly placed. I suggest it is moved to the methods section.

RESULTS

15. "Regression analyses revealed that the percentage of female authors, percentage of white authors, race of authors, gender of authors, and the number of authors is not correlated with the likelihood of being cited. Publication year was the only factor affecting to be cited"

Authors should support these findings with statistical inference. Which table or figure is being referred to in the work? Similar comment in the OLS regression.

16. I suggest authors go through the manuscript and correct sentences starting with numbers, eg, line 333

LIMITATION

17. Is the last paragraph part of limitations of the study? if not, authors should consider giving it appropriate heading. Other headings can also be considered in the manuscript especially the methods section

6. PLOS authors have the option to publish the peer review history of their article (what does this mean?). If published, this will include your full peer review and any attached files.

**Do you want your identity to be public for this peer review?** For information about this choice, including consent withdrawal, please see our Privacy Policy.

Reviewer #1: **Yes**

Reviewer #2: No

---

## [Decision Letter · Decision Letter 1]

11 Dec 2023

A scientometric analysis of fairness in health AI literature

PGPH-D-23-00817R1

Dear Dr. Quion,

We are pleased to inform you that your manuscript 'A scientometric analysis of fairness in health AI literature' has been provisionally accepted for publication in PLOS Global Public Health.

Best regards,

Zahra Zeinali, MD MPH DrGH (c)

Academic Editor

Reviewer Comments (if any, and for reference):

Reviewer's Responses to Questions

**Comments to the Author**

1. If the authors have adequately addressed your comments raised in a previous round of review and you feel that this manuscript is now acceptable for publication, you may indicate that here to bypass the “Comments to the Author” section, enter your conflict of interest statement in the “Confidential to Editor” section, and submit your "Accept" recommendation.

Reviewer #1: All comments have been addressed

Reviewer #2: All comments have been addressed

2. Does this manuscript meet PLOS Global Public Health’s publication criteria? Is the manuscript technically sound, and do the data support the conclusions? The manuscript must describe methodologically and ethically rigorous research with conclusions that are appropriately drawn based on the data presented.

Reviewer #1: (No Response)

Reviewer #2: Yes

3. Has the statistical analysis been performed appropriately and rigorously?

Reviewer #1: Yes

Reviewer #2: Yes

4. Have the authors made all data underlying the findings in their manuscript fully available (please refer to the Data Availability Statement at the start of the manuscript PDF file)?

Reviewer #1: Yes

Reviewer #2: Yes

5. Is the manuscript presented in an intelligible fashion and written in standard English?

Reviewer #1: Yes

Reviewer #2: Yes

6. Review Comments to the Author

Reviewer #1: Thank you for thoroughly addressing the comments made on the original manuscript.

Reviewer #2: (No Response)

7. PLOS authors have the option to publish the peer review history of their article (what does this mean?). If published, this will include your full peer review and any attached files.

**Do you want your identity to be public for this peer review?** For information about this choice, including consent withdrawal, please see our Privacy Policy.

Reviewer #1: No

Reviewer #2: No
